# The Prevalence and Risk Factors of Low Bone Mineral Density in the Population of the Abay Region of Kazakhstan

**DOI:** 10.3390/ijerph21060681

**Published:** 2024-05-26

**Authors:** Madina Madiyeva, Tamara Rymbayeva, Alida Kaskabayeva, Gulzhan Bersimbekova, Gulnur Kanapiyanova, Mariya Prilutskaya, Dinara Akhmetzhanova, Aliya Alimbayeva, Nazarbek Omarov

**Affiliations:** 1Department of Radiology, Semey Medical University, Abay Street, 103, Abay Region, Semey 071400, Kazakhstan; 2Department of Internal Diseases and Rheumatology, Semey Medical University, Abay Street, 103, Abay Region, Semey 071400, Kazakhstan; tamara.rymbayeva@smu.edu.kz (T.R.); alida.kaskabayeva@smu.edu.kz (A.K.); gulzhan.bersimbekova@smu.edu.kz (G.B.); gulnur.kanapiyanova@smu.edu.kz (G.K.); 3Department of Personalised Medicine, Pavlodar Branch of Semey Medical University, TorajgyrovStreet 72/1, Pavlodar Region, Pavlodar 140001, Kazakhstan; mariya.prilutskaya@smu.edu.kz; 4Department of Pediatrics and Medical Rehabilitation Named after Tusupova D.M., Semey Medical University, Abay Street, 103, Abay Region, Semey 071400, Kazakhstan; dinara.akhmetzhanova@smu.edu.kz (D.A.); aliya.alimbayeva@smu.edu.kz (A.A.); 5Scientific Research Department, Semey Medical University, Abay Street, 103, Abay Region, Semey 071400, Kazakhstan; nazarbek.omarov@smu.edu.kz

**Keywords:** bone mineral density, osteoporosis, dual-energy X-ray absorptiometry, dietary patterns, aging

## Abstract

Osteoporosis is considered a serious public health problem that particularly affects the postmenopausal period. In 2018, in the Republic of Kazakhstan, the prevalence of osteoporosis was 10.0, and the incidence was 3.7 new cases, per 100,000 adults, respectively. The objective of this study was to assess the prevalence of osteoporosis and indicate the main factors affecting low bone mineral density by screening the adult population of the Abay region, Kazakhstan. The target group comprised 641 respondents aged between 18 and 65 years old, from a Kazakh population, who had been living in the Abay region since birth. All participants filled out a questionnaire and were subjected to a bone mineral density measurement by means of dual-energy X-ray absorptiometry (DXA) between 15 July 2023 and 29 February 2024. Logistic regression analysis was conducted to assess the association between low bone mineral density and key demographic characteristics, such as lifestyle factors and nutritional habits. We identified the prevalence of low bone mass (osteopenia) and osteoporosis to be 34.1%, with the highest prevalence of 48.3% being found in the older population group (50+ years). The regression analysis revealed a number of indicators associated with the likelihood of bone sparing. However, only four of these showed significance in the final multivariate model (R^2^ = 22.4%). These were age (adjusted odds ratio (AOR) 1.05) and fracture history (AOR 1.64) directly associated with the likelihood of low bone density. Meanwhile, the body mass index (AOR 0.92) and the consumption of nuts and dried fruits (AOR 0.48) reduced the chance of bone tissue demineralization. Additional studies examining the prevalence and any emerging risk factors for osteoporosis are needed to advance clinical epidemiological knowledge and implement public health programs.

## 1. Introduction

Osteoporosis is a global health disease characterized by a reduction in bone mass and disruption in the microarchitecture of bone leading to an increased predisposition for fractures [1].

Osteoporosis is considered a serious public health problem and particularly affects the postmenopausal period, when there is a marked decrease in the production of estrogen. The diagnosis of osteoporosis can be made on the basis of either fractures that have occurred without significant trauma or low bone mineral density, measured by means of dual-energy X-ray absorptiometry (DXA), which is considered to be the “gold standard” in the diagnosis of osteoporosis. An osteoporotic fracture can occur in almost any part of the skeleton, with the spine, hip, wrist, humerus, and pelvis being the most commonly affected [2]. It is known that a 1.0SD decrease in bone mineral density (BMD) increases the risk of fractures by 2–2.5 times [3].

The prevalence of osteoporosis is continuing to escalate with the increasingly elderly population [4]. As the global population ages, the estimated annual number of hip fractures is expected to double between 2018 and 2050 [5]. On 1 February 2024, the population of Kazakhstan was 20,053,665 people, including 9,793,608 men and 10,260,057 women. The number of people who were 60 years old and older amounted to 2,616,515 people (13.2% of the total population), with 1,031,051 (39.4%) being men and 1,585,064 (60.6%) being women. It is projected that by 2035, the population over 50 years of age will grow by 35%, and that over 70 years of age will grow by 95%. In the Abay region of Kazakhstan, there are 610,100 people, including an adult population of 421,501 people [6]. Osteoporosis is still quite rarely registered in the official statistics of Kazakhstan. In 2018, 1245 cases of osteoporosis were registered in the Republic of Kazakhstan, with an expected number of 1.1 million patients. The prevalence of osteoporosis was 10.0, and the incidence was 3.7 new cases per 100,000 adults, respectively [7]. The incidence of osteoporotic fractureswas naturally predominant in women, with 236 cases as opposed to 181 cases in men (per 100,000 population), and it increased with age [8]. There areenough publications on osteoporosis and osteoporotic fractures in groups of people over 65 years of age [8]. We found no studies in Kazakhstan on the prevalence of osteoporosis in young and middle-aged people using dual-energy X-ray absorptiometry.

Osteoporosis has many causes, including age, genetic factors, hormone therapy, some somatic diseases, prolonged bed rest, low physical activity, and nutritional status [9]. Malnutrition and a lack of sunlight may also be responsible for decreased bone mineralization. An adequate intake of selected nutrients that are rich in calcium, vitamin D, n-3polyunsaturated fatty acids (n-3 PUFAs), and protein-rich foods is essential for healthy bones [10]. Most people are asymptomatic with osteoporosis, making epidemiologic studies particularly difficult. Osteoporosis can be prevented and treated. It is therefore important that risk factors are identified and continually updated to ensure that preventive care is as complete as possible. The objective of this study was to assess the prevalence of osteoporosis and determine the main factors affecting low bone mineral density by screening the adult population aged 18 to 65 years of the Abay region, Kazakhstan.

## 2. Materials and Methods

### 2.1. Design

The study recruitment was conducted by means of the random sampling method using random number tables and the register of the target population. The target group was people aged 18–65 years living in the Abay region of Kazakhstan since birth.

The study was approved by the Ethics Committee of Semey Medical University (SMU) (Protocol No. 7 dated 7 November 2022). All the participants filled out questionnaires and had their BMD measured by means of DXA between 15 July 2023 and 29 February 2024. Each participant’s survey data were compared with their bone density scans at the University Hospital of the SMU and “Toktamys” Medical Center. This study was pre-registered in the international clinical trials registry on the website Clinicaltrials.gov; accessed on 27 March 2024 (ID NCT06344598).

### 2.2. Subjects

The subjects of the study were 846 residents of the Abay region of Kazakh nationality. Of these, 641 respondents (564 female and 77 male) were included in the sample according to the inclusion and exclusion criteria. In our study, the number of women exceeded the number of men. In fact, only a small number of men took part in the study and underwent DXA screening. This is consistent with the study by Rinonapoli G. et al. [11].

The inclusion criteria included adults over 18 years old without a congenital pathology of the musculoskeletal system and without any disorders affecting bone metabolism, who had been living in the Abay region since birth, and who were not in the acute phase of a somatic pathology. The exclusion criteria included being under 18 years and over 65 years of age, patients with a congenital pathology of the musculoskeletal system, residents of other regions of Kazakhstan, or being unwilling to participate in the study and unable to sign the informed consent independently. The problem of the insufficient availability of diagnostic equipment remains urgent in Kazakhstan [7]. The bone densitometer is located in the city of Semey (the administrative center of Abay region) and is accessible to residents of this region but was technically inaccessible to residents of other regions of Kazakhstan due to large distances between regions. The present study is the initial stage of a grant project, which is scheduled to be carried out in 2023–2025.

### 2.3. Bone Measurements and the Survey

The subjects’ BMDs were measured using DXA (Osteosys, 2020, Seoul, Republic of Korea). We measured the bone density of the lumbar spine (L1–L4) in accordance with the guidelines for the examination of bone density based on the recommendation of the International Society for Clinical Densitometry (ISCD) from 2007 [12]. The bone region of interest was determined manually and sometimes automatically.

Osteoporosis is defined based on the following bone density levels: T-scores and Z-scores of −2.5 SD and below indicate the presence of osteoporosis; from −1.5 SD to −2.5 SD, they indicate a low bone mass (osteopenia), and a value equal to or exceeding −1.4 SD is considered to indicate normal bone density [2]. The values of bone mineral density were interpreted by a qualified radiologist.

The questionnaires were developed in accordance with the international questionnaire for the diagnosis of osteoporosis, which is available for free at IOF www.osteoporosis.foundation (accessed 9 April 2024) [13], and supplemented with questions to obtain demographic information, including the subjects’ date of birth, place of birth and residence, and nationality. To measure the validity of the questionnaires, the same respondents were retested and the percentage of discrepancies in responses was assessed. In total, validation was carried out on 30 questionnaires. The reliability of the questionnaires was assessed by analyzing internal consistency, and content validity was assessed at the questionnaire development stage. Specific risk factors for osteoporosis that were assessed in the questionnaire included the height and body weight with a BMI calculation, history of smoking and alcohol consumption, use of hormone therapy, immunosuppressants, antacids, antidiabetic agents, previous fractures, family medical history, diet, and physical activity. The patients were asked about the presence of chronic diseases affecting osteoporosis, including endocrine, rheumatic, and cancer pathologies. Participants were also asked about their current calcium and vitamin D intake. Nutrition questions included “how often/rarely consumed” foods that are rich in calcium and protein—such as milk and dairy products, red meat and meat products, fish and seafood, different types of nuts and dried fruits, and vegetables—were. Questions on physical activity included the time spent outdoors and physical activity during the day. People who agreed to participate in the study were surveyed in a paper format, signing an informed consent form and providing a contact telephone number.

### 2.4. Analysis Method

Study participants were divided into 2 groups according to age: participants who were younger than 50 years and those older than 50 years. Statistics show that the mean age of natural menopause in industrialized nations is 51 years [14]. The prevalence of osteoporosis over the age of 50 years is 7% in men, which is lower than the 23% reported for women [15]. Therefore, there was no sex difference when dividing the participants into two groups, as the prevalence amounted to 12% in men and 88% in women. The inclusion of respondents under 50 years of age in the study was dictated by a scientific interest in the prevalence of osteoporosis in this group of the population of the Abay region of Kazakhstan.

The frequencies were compared using either Pearson’s chi-square test or Fisher’s exact test (applied when expected cell values were five or less). The continuous variables were compared using the Mann–Whitney U-test due to their non-normal distribution, which was confirmed by means of the Shapiro–Wilk test. Logistic regression analyses were conducted to assess the association between osteopenia and osteoporosis and key demographic characteristics, the aforementioned lifestyle factors, and eating habits. In multivariate analyses, the unadjusted and adjusted odds ratios were presented. The model’s effect size was determined by calculating the Nagelkerke R^2^. The analyses were performed using IBM SPSS version 22 (IBM Corp., Armonk, NY, USA)

## 3. Results

The final sample amounted to 641 respondents. The number of women was 564 (88.0%), and the number of men was 77 (12.0%). In the less than 50 years group there were 266 women and 44 men. In the group of 50 years and above, there were 238 women and 33 men. No differences in sex distribution between the age groups were found.

The mean BMI was 24.2 (7.05) kg/m^2^. In patients who were below 50 years of age, the mean BMI was 23.2 (6.12) kg/m^2^, and in patients who were 50 years and older, the mean BMI was 25.3 (7.5) kg/m^2^. According to the results of the questionnaire, 29 (4.5%) respondents indicated that they smoke cigarettes, and 11 (1.7%) people indicated an unhealthy habit in the form of alcoholic beverage intake. A history of a use of hormonal therapy (past or present) was indicated by 74 (11.5%) people. Regarding fractures after minor injuries and falls, these were reported by 112 (33.8%) respondents from the age group of over 50 years out of the 158 people in total who reported such injuries. The BMD measurements of all respondents showed the following results: 20.2% had a low bone mass (osteopenia), 13.9% had osteoporosis, and 65.9% had normal BMD (Figure 1). The highest BMD reduction was observed in the group who were 50 years and older (Table 1 and Table 2). The results of the nutrition preference questionnaire in the two age groups are presented in Table 3.


ijerph-21-00681-t004_Table 4Table 4Regression analysis of risk factors indicating a significant association with osteoporosis.ParameterOR95% CI
*p*
AOR95% CI
*p*
Age1.051.04; 1.06<0.0011.051.04; 1.06<0.001Sex


---Male0.770.45; 1.290.32


Femaleref




Weight (kg)0.980.97; 0.990.005---Height0.970.95; 0.990.003---BMI0.950.92; 0.980.0020.920.88; 0.95<0.001Chronic diseases 1.561.08; 2.260.0190.870.48; 1.570.64Rheumatoid arthritis2.131.31; 3.470.0021.710.85; 3.480.14Glucocorticoid (GC) consumption1.661.01; 2.710.041.020.52; 1.990.95History of fractures 2.251.56; 3.26<0.0011.641.07; 2.530.02Frequent falls or fear of falling1.691.15; 2.480.0080.980.61; 1.570.92Decrease in height 1.79 1.15; 2.790.0101.100.66; 1.840.71Insufficient physical activity0.920.64; 1.330.669---Alcohol consumption3.471.01; 11.980.0493.250.88; 12.030.08Lack of outdoor time0.820.52; 1.270.363---Lack of vitamin D consumption0.950.60; 1.500.82---Lack of calcium consumption0.730.42; 1.260.27---Cigarettes1.390.5; 2.960.40---Dairy products1.290.66; 2.520.50---Greens 1.210.82; 1.810.35---Meat1.560.31; 7.780.59---Fish0.880.51; 1.530.66---Consumption of nuts and dried fruits0.460.27; 0.780.0040.480.27; 0.850.012Eggs 0.690.39; 1.190.18---Soda 0.990.70; 1.410.97---Fast food0.890.64; 1.250.51---OR—odds ratio; AOR—adjusted odds ratio; 95%CI—confidence interval.


## 4. Discussion

To the best of our knowledge, our study was the first in Kazakhstan to investigate the association of key demographic, behavioral, and anamnestic factors with low bone mineral density. We identified the prevalence of low bone density to be 34.1%, with the highest prevalence of 48.3% being found in the older population group (50+ years). Besides age, low bone density was directly associated with a history of fractures. BMI and eating nuts and dried fruits were inversely associated with low bone mineral density.

Epidemiologically, osteoporosis predominantly occurs in postmenopausal and premenopausal women, as well as in men who are over 50 years of age. Risk factors affect different ages and are not definitive; thus, the incidence and risk factors of osteoporosis in the Kazakh population were studied in order to assess their impact on BMD. Risk factors were identified by means of a unified questionnaire and included age; anamnestic and nutritional factors; and vitamin D, calcium, and medication consumption.

Several studies have evaluated the risk of osteoporosis in postmenopausal women aged 45 years and older, and the positive association between osteoporosis and an age > 45 years [15,16,17]. In one study, osteoporosis was found in 12% of women aged 40–49 years, 21.8% of women aged 50–59 years, and 45.7% of women aged > 60 years [16]. In another observational study, the prevalence of osteoporosis was significantly lower in those aged 40–50 years than in those aged 50 years and older, by a factor of about 20–40 [17]. The mean age of women was 59.5 ± 8.6 years, and the mean age of menopause onset was 49.0 ± 3.4 years [18].

Our study was a screening study with respondents who were younger than 50 years and older than 50 years. It was found that with an increase in age of 1 year, the risk of osteoporosis increased by 5% (Table 4). We also found that 65.9% of the Kazakh population had normal BMD in the lumbar spine, 20.2% had osteopenia, and 13.9% had osteoporosis. In the group of 50 years and older, 57.1% had normal BMD in the lumbar spine, 26.0% had osteopenia and 22.4% had osteoporosis (Figure 1). The highest prevalence of reduced BMD was in individuals who were over 50 years of age, which is consistent with the findings of numerous studies reporting that individuals over 50 years of age are five times more likely to have osteoporosis than the general population [19,20].

When measuring the lumbar spine (LS) in Turkish women [21], the incidences of normal BMD (31.4%), osteopenia (48.2%), and osteoporosis (20.5%) were higher than those found in another study by the Turkish researchers İpek A et al. [18]. In a study by Thambiah SC et al. [22], the prevalence of osteoporosis in the LS in Thai women in the age range of 55–59 years was 22.6%.

In China, the age-standardized prevalences of osteoporosis in the spine or hip were 6.46% and 29.13% for men and women aged 50 years and older, respectively [23]. In the same study, the authors found that the prevalence of osteoporosis at each site increased with age in the range of 5 years, which may only be partially comparable to our study. In our study sample the BMD significantly decreased for each year of body aging (AOR 1.05; *p* < 0.001). We found no studies on the association of an increased risk of osteoporosis with an increasing age with each 12 months. A total of 158 individuals had chronic diseases, including 110 individuals (33.2%) in the age group who were older than 50 years. Pathologies such as thyroid and parathyroid gland diseases (11.5%, *p* < 0.001) and rheumatoid arthritis (16.9%, *p* < 0.001) were statistically significant, and accordingly, those as well as the GC intake (16.3%, *p* < 0.001) were the factors that increased the risk of osteoporosis in the group who were50 years old and older.

According to the Framingham study [24], individuals who generally followed a diet based on fruits, vegetables, milk, and cereals had significantly higher BMD than those whose diet was characterized by a high consumption of salty snacks, pizza, and sodas, or a high consumption of meat, bread, and potatoes. Seafood is known to be rich in n-3PUFAs such as eicosapentaenoic acid and docosahexaenoic acid, which inhibit the production of inflammatory cytokines, enhance calcium absorption, reduce urinary calcium excretion, and regulate bone health [10]. Epidemiologic studies have shown that fish consumption and n-3PUFAs were significantly associated with BMD, fractures, and osteoporosis risk in postmenopausal women or elderly men in an Asian population. However, n-3PUFAs or fish consumption was not found to be associated with BMD or fractures in elderly men and women in Western populations [25].

The relationship of diet to the bone health of the Kazakh population can be analyzed by focusing on selected nutrients such as meat, seafood, dairy products, nuts, and vegetables. According to our survey, 81.7% of the population rarely eats seafood, and 64.6% rarely eat nuts, while 53.0% of the respondents said that they do not eat greens often, regardless of age. Soda and fast food are more often consumed by those who are under 50 years of age compared to the group over 50 years of age, with 54.2% (*p* < 0.001) and 65.8% (*p* < 0.001), respectively, which are statistically significant (Table 3).

It is known that protein-rich foods from different sources can have different effects on bone health, as they vary in their protein content, amino acid composition, and digestibility [26]. The relationship between BMD and protein-rich foods was analyzed in 2015, and the authors found that the processed foods and red meat protein food clusters were related to lower bone mineral density [27]. The authors believe that this is due to the higher saturated fat content of red meat compared to other sources of animal protein. In another study, it was also found that a diet that is high in fish and olive oil and low in red meat was positively associated with the LS’s BMD [28]. The explanation for the beneficial effect of protein on bone formation in a balanced diet, in terms of the acid-forming potential, is that an acid/base balance is important to avoiding urinary calcium loss, and can be affected by acid-forming foods such as processed meat [29].

The Kazakh population is known for its preference for red meat and meat products, and that was found in this study. In our study, 88.8% (*p* = 0.04) of respondents consumed meat, without age differentiations. Possibly, this factor affects the prevalence of osteoporosis in the Kazakh population; however, no correlation was found (Table 4).

Thus, based on our data and the few studies regarding red meat consumption [28,29] and the risk of osteoporosis, we suggest that research on the various factors affecting bone health should take the ethnic and geographic population into account.

Other non-dietary factors also influence bone metabolism. First of all, motor activity is essential for skeletal muscles: physical activity helps maintain or build skeletal muscle volume and strength, and constant and individually adjusted physical activity strengthens bones at any age. Other epidemiologic studies have shown that a 10% increase in peak bone mass at the population level reduces the risk of fracture in later life by 50% [30,31].

In the survey, the questionnaires were formulated according to the IOF recommendations, such as the following question: “Do you engage in physical activity for more than 30 min per day (housework, gardening, walking, running)” [13]. In our study, 72.2% of the respondents gave a positive answer, and individuals in both groups were equally likely to engage in physical activity. However, we found no correlation with this risk factor.

In our study, respondents’ fracture history exhibited a positive correlation with the risk of osteoporosis. Previous publications have noted that the risk of subsequent fracture is approximately four times greater in women with one previous fracture [32]. We found a similar association between prior fractures and osteoporosis of 1.64 (*p* < 0.001).

There are now sufficient publications on the effect of the BMI on bone metabolism in postmenopausal women, and a significant association between the BMI and BMD at the lumbar and femur has been demonstrated [33]. We observed the effect of BMI on the development of osteoporosis by means of regression analysis. We found that a lower BMI increased the risk of osteoporosis by 8% (*p* < 0.001).

This study is limited by the fact that the sample included women and men 18–65years old living in the sharply continental climate of one region of Kazakhstan, and this means that the results may not be generalizable to all of Central Asia and require further study. In addition, data collection was prospective, and some data were retrospective, leading to inherent limitations. Of note, three variables were used as the main outcome variables: osteoporosis, osteopenia, and normal BMD, measured at the lumbar spine. Based on the findings that the risk of osteoporosis increases with increasing age by 5%, further research should be conducted on a larger sample with DXA measurements in two anatomical zones, including the femoral neck, with an aim to create an algorithm for predicting osteoporotic fractures in women and men living in Kazakhstan.

The main strength of our study is the assessment of the overall prevalence of osteoporosis in the adult population of different ages, who all live in one geographical area of Kazakhstan. Consequently, the screening enabled us to study risk factors across age ranges. The identified correlations might help in diagnosing osteoporosis in high-risk patients earlier, even before menopause. This knowledge will also help clinicians take appropriate treatment measures and recommend lifestyle and medication adjustments.

## 5. Conclusions

This study in the Abay region of Kazakhstan found that the prevalence of low bone mineral density was higher in individuals 50+ years of age than in individuals <50 years of age. Age, BMI, history of fractures, and insufficient dietary intake of calcium and protein are factors influencing bone mineralization in the population. It is necessary that the entire population of Kazakhstan undergo a comprehensive diagnosis of this disease. From a public health point of view, it is valuable to know about any new risk factors in order to evaluate these factors and build differentiated preventive policies for osteoporosis at different ages.

## Figures and Tables

**Figure 1 ijerph-21-00681-f001:**
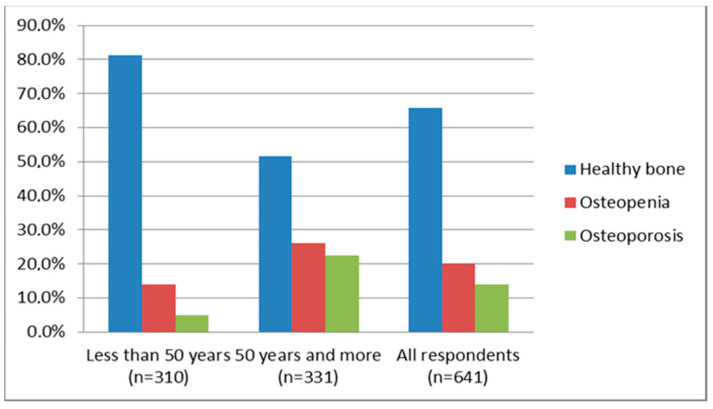
BMD in the population of the Abay region of Kazakhstan.

**Table 1 ijerph-21-00681-t001:** General findings on osteoporosis risk factors.

Parameter	All Respondents (*n* = 641)	Less than 50 Years (*n* = 310)	50 Years and Above (*n* = 331)	Statistical Criterion	*p*-Value
Female	564 (88.0%)	266 (85.8%)	298 (90.0%)	χ^2^ = 2.7	0.1
Male	77 (12.0%)	44 (14.2%)	33 (10.0%)		
Densitometry				χ^2^ = 67.94	<0.001
Healthy bone	422 (65.9%)	251 (81.2%)	171 (51.7%)		
Low bone mass (Osteopenia)	129 (20.2%)	43 (13.9%)	86 (26.0%)		
Osteoporosis	89 (13.9%)	15 (4.9%)	74 (22.4%)		
Fractures after minor injuries and falls	158 (24.6%)	46 (14.8%)	112 (33.8%)	χ^2^ = 31.11	<0.001
Frequent falls or fear of falling	140 (21.8%)	35 (11.3%)	105 (31.7%)	χ^2^ = 39.15	<0.001
After the age of 40, have you lost more than 3 cm in height	94 (14.7%)	23 (7.4%)	71 (21.5%)	χ^2^ = 25.18	<0.001
Chronic diseases	158 (24.6%)	48 (15.5%)	110 (33.2%)	χ^2^ =27.15	<0.001
Hepatitis	7 (1.1%)	4 (1.3%)	3 (0.9%)		0.72 *
Chronic obstructive pulmonary disease (COPD)	3 (0.5%)	1 (0.3%)	2 (0.6%)		1.0 *
Cancer	9 (1.4%)	4 (1.3%)	5 (1.5%)		1.0 *
Diabetes	15 (2.5%)	4 (1.3%)	12 (3.6%)		0.08 *
Thyroid or parathyroid gland disorders	60 (9.4%)	22 (7.1%)	38 (11.5%)	χ^2^ = 3.63	0.06
Rheumatoid arthritis	74 (11.5%)	18 (5.8%)	56 (16.9%)	19.36	<0.001
Drug therapy					
Antidiabetic	10 (1.6%)	2 (0.6%)	8 (2.4%)		0.11 *
Antacids	2 (0.3%)	1 (0.3%)	1 (0.3%)		1.0 *
Immunosuppressants	7 (1.1%)	0	7 (2.1%)		0.02 *
Glucocorticoids	74 (11.5%)	20 (6.5%)	54 (16.3%)	χ^2^ = 15.25	<0.001
Vitamin D	94 (14.7%)	40 (12.9%)	54 (16.3%)	χ^2^ = 1.49	0.22
Calcium	59 (9.2%)	22 (7.1%)	37 (11.2%)	χ^2^ = 3.19	0.07

*—Fisher’s exact test.

**Table 2 ijerph-21-00681-t002:** Behavioral risk factors for osteoporosis development.

Parameter	All Respondents (*n* = 641)	Less than 50 Years (*n* = 310)	50 Years and Above (*n* = 331)	Statistical Criterion	*p*-Value
Physical activity	463 (72.2%)	225 (72.6%)	238 (71.9%)	χ^2^ = 0.04 ^a^	0.85
Being outdoors	529 (82.5%)	262 (84.5%)	267 (80.7%)	χ^2^ = 1.65	0.19
Family history *of* osteoporosis	87 (13.6%)	39 (12.6%)	48 (14.5%)	χ^2^ = 0.51	0.48
Parents’ history of fractures	87 (13.6%)	45 (14.5%)	42 (12.7%)	χ^2^ = 0.46	0.50
Alcohol, 3 or more units/day	11 (1.7%)	3 (1.0%)	8 (2.4%)		0.23 *
Current smoking	29 (4.5%)	15 (4.8%)	14 (4.2%)	χ^2^ = 0.14	0.71
Weight, kg	67.0 (20.0)	63.0 (16.5)	70.0 (18.0)	U = 39,371.0 **	<0.001
Height, cm	162.0 (10.5)	163.0 (10.3)	160.0 (9.0)	U = 43,943.5 **	0.002
BMI, kg/m^2^	24.2 (7.05)	23.2 (6.12)	25.3 (7.5)	U = 43,272.0 **	0.001

**—nonparametric criteria—Mann–Whitney U-test; a—χ^2^ Pearson’s; *—Fisher’s exact test.

**Table 3 ijerph-21-00681-t003:** Results of patient survey on nutrition.

Parameter/Frequency of Use	All Respondents (*n* =641)	Less than 50 Years (*n* =310)	50 Years and Above (*n* =331)	Statistical Criterion	*p*-Value
Consumption of milk and dairy products				χ^2^ = 0.48	0.79
none	45 (7.0%)	24 (7.7%)	21 (6.3%)		
rarely	326 (50.9%)	156 (50.3%)	170 (51.4%)		
often	270 (42.1%)	130 (41.9%)	140 (42.3%)		
Vegetables and greens				χ^2^ = 7.42	0.02
none	143 (22.3%)	79 (25.5%)	64 (19.3%)		
rarely	340 (53.0%)	168 (54.2%)	172 (52.0%)		
often	158 (24.6%)	63 (20.3%)	95 (28.7%)		
Meat products (red meat)				χ^2^ = 6.47	0.04
none	8 (1.2%)	7 (2.3%)	1 (0.3%)		
rarely	64 (10.0%)	26 (8.4%)	38 (11.5%)		
often	569 (88.8%)	277 (89.4%)	292 (88.2%)		
Fish and seafood				χ^2^ = 0.53	0.29
none	60 (9.4%)	33 (10.6%)	27 (8.2)		
rarely	524 (81.7%)	254 (81.9%)	270 (81.6%)		
often	57 (8.9%)	23 (7.4%)	34 (10.3%)		
Nuts and dried fruits				χ^2^ = 2.47	0.29
none	63 (9.8%)	26 (8.4%)	37 (11.2%)		
rarely	414 (64.6%)	209 (67.4%)	205 (61.9%)		
often	164 (25.6%)	75 (24.2%)	89 (26.9%)		
Eggs				χ^2^ = 1.78	0.41
none	57 (8.9%)	25 (8.1%)	32 (9.7%)		
rarely	372 (58.0%)	175 (56.5%)	197 (59.5%)		
often	212 (33.1%)	110 (35.5%)	102 (30.8%)		
Soda				χ^2^ = 51.99	<0.001
none	203 (31.7%)	62 (20.0%)	141 (42.6%)		
rarely	327 (51.0%)	168 (54.2%)	159 (48.0%)		
often	111 (17.3%)	80 (25.8%)	31 (9.4%)		
Fast food				χ^2^ = 76.30	<0.001
none	253 (39.5%)	72 (23.2%)	181 (54.7%)		
rarely	348 (54.3%)	204 (65.8%)	144 (43.5%)		
often	40 (6.2%)	34 (11.0%)	6 (1.8%)		

The regression analysis revealed a number of indicators associated with the likelihood of bone sparing. However, only four of these showed significance in the final multivariate model (Nagelkerke R^2^ = 22.4%). These included age and fracture history, which were directly associated with the likelihood of low bone density. Meanwhile, the body mass index and consumption of nuts and dried fruits reduced the chance of bone tissue demineralization (Table 4). The classification rate for the regressive model was 73.6%.

## Data Availability

The data necessary to reproduce the results presented here are not publicly accessible as the participants’ informed consent did not include public data sharing, but are available from the first author upon reasonable request.

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
