# Peer review of "The Prevalence and Risk Factors of Low Bone Mineral Density in the Population of the Abay Region of Kazakhstan"

_ijerph, 2024, doi:10.3390/ijerph21060681_

Round 1
Reviewer 1 Report
Comments and Suggestions for Authors
Dear Authors,
The manuscript titled ” The Prevalence and Risk Factors of Low Bone Mineral Density in the Population of the Abay region of Kazakhstan” is written in a professional manner, making it accessible to a wide scientific audience. While the topic of the manuscript could be very important, the introduction/discussion of the manuscript could have more explicitly stated how this research is different from other studies and why is of imprtance. The manuscript falls short in evaluating its own impact and novelty within the research field. Also, the wording when describing BMD measurement must include the skeletal site that was analyzed. The explanation of why the authors chose just this specific region of the country is not clearly explained. The methods are described with a reasonable level of detail, providing a foundation for replication. However, additional information related to some specific methodological details might be beneficial for readers aiming to replicate the study and interpret the data. Namely, it is of out most importance to state which skeletal site was measured and to alwaas refer this in the results section. Also, how the measurement was conducted is stated very unspecifically (“as standardized method – line 103”). The reader must know which level of the lumbar spine was analyzed (L1-L4 or L2-L4) and how the bone region of interest was determined (was it manually adopted, or was it always automatic). Information about the difference in the distribution of sex participants is not explanatory to readers as is stated in the manuscript (lines 99-101): “The present study is the initial stage of a grant project and planned to be carried out in 2023-2025. The results presented in the study are the initial stage of the grant. This explains sampling bias towards women.” This must be better addressed since the sex specificity of the data is not presented, and previous studies indicated its clinical relevance. It is not clear why only one region of the country was selected and why individuals older than 65, in whom osteoporosis and bone fractures are more prevalent, were excluded from the study. This should also be added to limitations. It is not stated whether the questionnaires were validated and what was the process of validation.
Information about the number of study participants is contradictory or includes typing errors (641- as stated in line 92 or 461 – as stated in line 146). The number of individuals per age group should be added to the text as well. A graphic representation of BMD plotting by age would be interesting to address the data in more detail. It is not clear how the author addressed the following stated in lines 131-132 of the manuscript – “However, both males and females over the age of 50 years have trabecular bone loss throughout their later life.” Please avoid over-statements that are not directly backed by the generated data.
Also, detailed data about conjoined diseases and the use of bone-affecting therapy must be included in the manuscript (or its supplementary files) to provide detailed insight into the studied sample. Also, the number of individuals per group should be stated. For example, it is crucial to add information about the use of antiresorptive therapy in your sample (bisphosphonate or other).
It is not clear what is meant by the statement that 20.2% of the participants had “low bone mass” (stated in line 157), so please be more specific in word choice.
My major concern is that the study's conclusions are not well-formulated. In the current version of the manuscript and abstract, the conclusion is not directly tied to the used methodology or to the generated data. Namely, it is stated that “the study was the first in Kazakhstan that investigates the association of key demographic, behavior, and anamnestic factors with low mineral density.” So, it must be improved significantly to reflect specific data generated.
Also, the references must be improved, since a lot of information that is not a direct result of this study is given without the references. This is quite important in the introduction/discussion section, which must be significantly improved. The definition of osteoporosis used in the introduction is outdated. Data about population (lines 56-60) or specific relation between BMD loss and fractures (lines 52-53) require references. Also, a statement in line 212, “In the Kazakh study sample, BMD significantly decreased for each year of body aging.” must include references.
I recommend a thorough revision of the manuscript using online tools or assistance from a fluent English speaker. The formatting should also be checked (a different font is used).
Comments on the Quality of English LanguageExtensive editing of English language required
Author Response
Thank you very much for taking the time to review this manuscript. Please find the detailed responses below and the corresponding revisions/corrections highlighted/in track changes in the re-submitted files.
Our responses is in the attached file

Reviewer 2 Report
Comments and Suggestions for Authors
I reviewed the article by Madiyeva and colleagues titled “The Prevalence and Risk Factors of Low Bone Mineral Density in the Population of the Abay region of Kazakhstan”. The manuscript aims to study osteoporosis prevalence and indicate the main factors affecting low bone mineral density by screening the adult population of the Abay region, Kazakhstan. However, below I present some considerations about the manuscript.
I recommend inserting an effect size analysis and classification into logistic regression.
I recommend presenting a clinical significance for the logistic regression results to make it easier for the reader to understand.
I recommend removing duplicate results; they are presented in the text and tables, and once presented, it is enough.
Specific comments
Abstract
Line 30: I suggest checking if the “-” symbol is to identify a negative direction in the 48.3%.
Line 32: I recommend presenting the meaning of the acronym “AOR”.
1. Introduction
Line 53: I recommend inserting the reference for this information.
Line 65: in the sentence “The total and primary incidence was 10 and 3.7 cases per 100,000 adults, respectively.” I didn't understand what the incidents were. Please explain this better.
Lines 68 – 78: I recommend putting everything in one paragraph.
2. Materials and Methods
2.3. Bone measurements and the survey
Lines 103 – 110: I recommend presenting the body site (e.g. femoral neck, lumbar spine) used to calculate the T- and Z-score.
3. Results
Line 147: I recommend removing the symbol (-) between men - 77. This gives the effect of a negative value. I suggest checking this throughout the manuscript. I also suggest checking this in every manuscript.
Tables: I recommend including the meanings of the acronyms in the legend (e.g. OR, AOR).
Author Response

(The authors gave the same response as above.)

Round 2
Reviewer 1 Report
Comments and Suggestions for Authors
Dear authors,
thank you for addressing my comments. I have no further suggestions.
Kind regards